# Mediator subunit Med15 dictates the conserved "fuzzy" binding mechanism of yeast transcription activators Gal4 and Gcn4

Lisa M. Tuttle [1,2], Derek Pacheco[1,4], Linda Warfield[1], Damien B. Wilburn[3], Steven Hahn [1✉] &
Rachel E. Klevit [2✉]

The acidic activation domain (AD) of yeast transcription factor Gal4 plays a dual role in transcription repression and activation through binding to Gal80 repressor and Mediator subunit Med15. The activation function of Gal4 arises from two hydrophobic regions within the 40-residue AD. We show by NMR that each AD region binds the Mediator subunit Med15 using a "fuzzy" protein interface. Remarkably, comparison of chemical shift perturbations shows that Gal4 and Gcn4, two intrinsically disordered ADs of different sequence, interact nearly identically with Med15. The finding that two ADs of different sequence use an identical fuzzy binding mechanism shows a common sequence-independent mechanism for AD-Mediator binding, similar to interactions within a hydrophobic cloud. In contrast, the same region of Gal4 AD interacts strongly with Gal80 via a distinct structured complex, implying that the structured binding partner of an intrinsically disordered protein dictates the type of protein–protein interaction.

[1] Division of Basic Sciences, Fred Hutchinson Cancer Research Center, Seattle, WA, USA. [2] Department of Biochemistry, University of Washington, Seattle, WA, USA. [3] Department of Genome Sciences, University of Washington, Seattle, WA, USA. [4] Present address: Inventprise LLC, Redmond, WA, USA. ✉email: shahn@fredhutch.org; klevit@u.washington.edu

The regulation of yeast Gal4-dependent genes is a classic model system for the study of eukaryotic gene activation[1,2]. In the absence of galactose, Gal4 activation function is blocked by the repressor Gal80, while in the presence of galactose, Gal4 activates the transcription of genes required for galactose catabolism. Gal4 consists of an N-terminal DNA-binding domain that targets the upstream activating sequence of Gal4-regulated genes, a large central domain that likely plays a regulatory role, and a C-terminal transcription activation domain (AD)[3–7]. Under non-inducing conditions, Gal80 binds the Gal4 AD tightly and inhibits its function[8,9]. Upon binding galactose and ATP, the transducer protein Gal3 binds Gal80 to inhibit repressor function, allowing the Gal4 AD to activate transcription[10–12]. Gal4 was the first transcription factor shown to function in yeast, plants, insects, and animals, demonstrating wide conservation of activation mechanisms across eukaryotes[13–16].

The Gal4 AD, residues 840–881, is conserved among yeasts, is predicted to be intrinsically disordered[17], and contains several short hydrophobic stretches set in a background of acidic residues (Supplementary Fig. 1). Within the AD, residues 855–870 are the most highly conserved and play direct roles in both activation and repression (through binding to Gal80 repressor). Despite this dual function, mutagenesis studies of Gal4 have revealed a fundamental difference in the requirements for repression and transcription activation. While individual mutations in five Gal4 residues within the AD disrupt Gal80 repression, no single residue is critical for activation function as mutation of nearly every residue within the AD is tolerated[18,19]. Furthermore, the Gal4–Gal80 interaction is high affinity ($K_d$ in the nM range[6]) and the Gal4 AD peptide binds as an α-helix within a cleft formed by the N- and C-terminal Gal80 domains[20,21]. Altogether, the existing information suggests that Gal4–Gal80 binding involves a sequence-specific interface, while AD function does not.

The Med15 subunit of the transcription coactivator Mediator is important for activation of Gal4 target genes and binding of Gal4 to Mediator requires the Med15 subunit[22,23]. Crosslinking studies showed that the Gal4 AD interacts with Med15 during transcription activation and biochemical analysis identified the N-terminus of Med15 as the Gal4-binding region[22,24]. How Gal4–Med15 binding occurs in the apparent absence of sequence-specific interactions and what properties allow the same region of Gal4 to bind different proteins through fundamentally different binding modes remain unanswered questions.

Acidic ADs are found in a large class of transcription factors, many of which regulate inducible genes[1,25]. Acidic ADs are thought to be intrinsically disordered, have functionally important hydrophobic residues embedded within regions of net negative charge[26], and often target multiple unrelated subunits of transcription coactivators, such as Mediator, SAGA, Swi/Snf, and others. The best studied activator of this class is yeast Gcn4, which contains tandem ADs that, like Gal4, bind to Med15[27–30]. The activating function of the Gcn4 C-terminal AD (cAD) is largely contained within a short 5-residue-long sequence embedded within an acidic background, while the N-terminal AD (nAD) consists of 4 hydrophobic segments, distributed over 100 residues within a disordered acidic polypeptide[27,31–34]. Each Gcn4 AD can bind any of several activator-binding domains (ABDs) within Med15 via a fuzzy interface, in which interactions are highly dynamic and no unique protein–protein interface exists. In the large complex of tandem Gcn4 ADs and the complete activator-binding region of Med15, fuzzy binding is retained with nearly every possible AD–ABD interaction detected in a mechanism termed a "fuzzy free-for-all"[32]. Whether the fuzzy-free-for-all mechanism is employed more generally by other transcription factors is an important open question.

We have investigated the interactions between Gal4 AD and Med15 using a combination of biochemical, molecular genetics, and nuclear magnetic resonance (NMR) analyses and find that Gal4 AD–Med15 interactions involve a dynamic fuzzy-binding mechanism. Despite the many sequence and mechanistic differences involved in Gal4 regulation, we find remarkable similarity in the interfaces of the Med15 ABDs with Gcn4 and Gal4, implying conservation of a sequence-independent interaction that involves a "cloud" of hydrophobic character. By extrapolation, our results suggest that a fuzzy-binding mechanism is conserved among many acidic activators and provide a rationale for how multiple transcription activators with different primary sequences can target the same set of ABDs.

## Results

**Multiple hydrophobic regions of the Gal4 AD are important for transcription activation.** The Gal4 AD (C-terminal residues 840–881) contains short hydrophobic segments embedded within an acidic background: defined as Region 1 (residues 840–851) and Region 2 (residues 855–869), where each contains at least three aromatic residues (Trp, Tyr, or Phe) (Fig. 1 and Supplementary Fig. 1). A second segment with AD potential (Gal4 residues 148–196) is situated near the DNA-binding domain but does not appear to function in the context of full-length Gal4[5,26]. As there is some controversy regarding which residues contribute to Gal4 AD function[35], we first identified residues required for activation in vivo. We used a model system, in which Gal4 residues 828–881 were fused to the N-terminal linker region and DNA-binding domain of Gcn4 (residues 132–281). This Gcn4 derivative has no inherent AD function and can accept a wide variety of natural and synthetic ADs to allow activation of yeast Gcn4-dependent genes[31,36]. We used a Gal4 AD segment longer than the minimal AD because initial studies indicated that residues 840–881 formed a hydrogel upon purification while residues 828–881 remain soluble at very high concentrations, permitting structural, biochemical, and molecular analyses using the same Gal4 AD residues. Gal4 AD function was measured at the Gcn4-dependent genes *ARG3* and *HIS4* after addition of sulfometuron methyl (SM) to induce amino acid starvation and induction of Gal4-Gcn4 protein synthesis (see Supplementary Data 1).

Deletion of either the first 12 or last 10 residues of the 828–881 polypeptide had little or no effect on AD function, identifying the

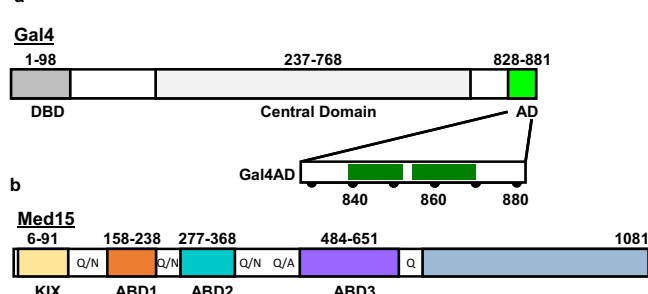

**Fig. 1 Gal4 AD is an IDR that interacts with Gal15 ABDs. a** Gal4 consists of a DNA binding and dimerization domain (DBD), a central domain (CD), and a primary activator domain (AD). The expanded view of the AD highlights key hydrophobic Regions 1 and 2 (residues 840–851 and 855–869, respectively). The DBD is structured upon DNA binding; the CD is sequence predicted to be predominantly helical; the AD is intrinsically disordered and includes a Gal80 binding site. **b** Med15 consists of four helical activator-binding domains (ABDs) and a C-terminal domain (blue) that is required for incorporation into the Mediator complex. The structured Med15 domains are connected by Q-rich linkers as shown.

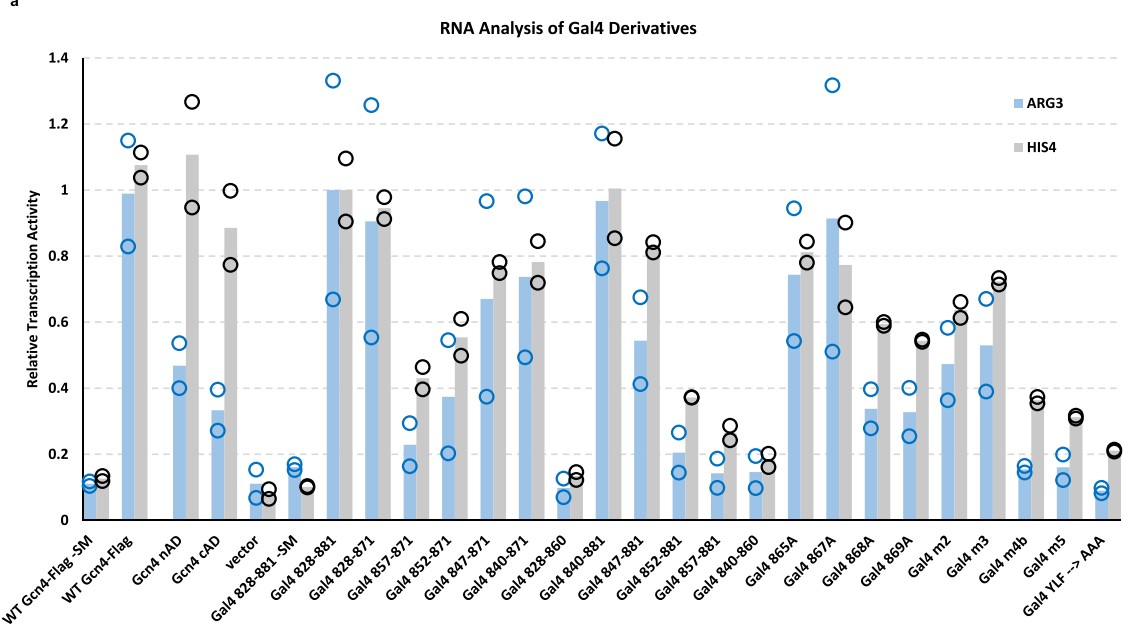

**Fig. 2 Residues throughout Gal4 840–871 positively contribute to transcription activation. a** RNA analysis of Gal4 AD-Gcn4 DBD derivatives. RNA levels measured by RT qPCR are normalized to *ACT1* mRNA and Gal4 828–881 + SM is set to 1; data points are shown as open circles and the average value is shown as bars. **b** Sequence of Gal4-Gcn4 derivatives and summary of activation assays from **a** at *ARG3*. Gal4 AD hydrophobic patches (green brackets above the sequence) and alanine mutations within these regions (red residues) are shown. See also Supplementary Fig. 1.

minimal Gal4 AD as residues 840–871 (Fig. 2). The minimal AD is composed of two fairly short regions, termed Region 1 and Region 2, that each contains large hydrophobic/aromatic residues. Truncations from residue 840 onwards that eliminate part or all of Region 1 reduce transcription activity by ~2–5-fold (847–881 and 852–881, respectively). Truncation from the C-terminal end of the Gal4 AD to eliminate most of Region 2 (840–860) reduced transcription activity by ~7-fold. Thus, the two individual hydrophobic regions of Gal4 have little function on their own but synergize when combined. Our analysis did not reveal the proposed self-inhibitory region located within the

AD (residues 872–881)[35] as removal of these residues (i.e., the construct 828–871) was not associated with an increase in measured activity.

Consistent with the importance of hydrophobicity, substitution of the three aromatic residues in Region 1 to alanine is nearly equivalent to a construct lacking Region 1 ("m4b" in Fig. 2). Region 2 contains four aromatic residues, Phe856, Tyr865, Tyr867, and Phe869. With the exception of Phe856, the aromatics are contained within a sequence ($^{865}$YNYLF) that resembles a short sequence motif in the Gcn4 central AD that is critical for function (WXXLF)[31] (Supplementary Fig. 1b). Despite this

similarity, we previously found that shortening Gal4 AD to contain only this segment (857–881) resulted in a construct with little inherent function (see Fig. S1b, c in ref. [31]). Individual alanine substitution of residues in Region 2 has differential effects: replacement of Leu868 or Phe869 (equivalent of the LF in the Gcn4 cAD motif; 0.34 ± 0.06 and 0.33 ± 0.07 for *ARG3*, respectively) reduced activity by threefold, while replacement of Tyr865 or Tyr867 has more modest effects (0.74 ± 0.20 and 0.91 ± 0.40 for *ARG3*, respectively). Notably, simultaneous replacement of the three positions that define the motif (i.e., Tyr867, Leu868, and Phe869; "YLF-to-AAA") produces a construct with at least tenfold decreased activation of *ARG3*. Altogether, our analysis identifies a minimal Gal4 AD (residues 840–871) composed of two hydrophobic regions that contribute to transcriptional activation of *ARG3*. Notably, Region 2 is essentially the same sequence shown to bind to the repressor protein Gal80 with high affinity[19].

**Gal4 AD interacts with Med15 ABDs**. Transcriptional activation by Gal4 requires its direct interaction with the Med15 subunit of the Mediator complex. Med15 contains four ABDs, known as ABD1, ABD2, ABD3, and KIX (Fig. 1b)[29,30]. We measured binding affinities of Gal4 AD (the construct containing Gal4 residues 828–881 was used for all binding and NMR experiments) with each individual Med15 ABD, KIX, and with a combined ABD123 that contains the three Med15 ABDs connected by shortened linkers. The length of the linkers between the ABDs is variable across yeasts[37,38]. We chose to use shortened linkers to be consistent with a construct used in previous experiments[32]. We expect that linker length and content may affect relative binding affinities when comparing individual ABD binding to the whole but also note that the full extent of the role of these linkers in vivo is not yet known.

Fluorescence polarization (FP) and/or isothermal calorimetry (ITC) measurements reveal that Gal4 AD binds each ABD with micromolar affinity (Table 1), with strongest binding to ABD1 ($K_d$ of 9 μM), followed by ABD3 ($K_d$ of 16 μM) and ABD2 ($K_d$ of 35 μM). Binding is enhanced in the presence of all three ABDs, with a binding constant value of 2 μM for ABD123. Binding of Gal4 AD to KIX was too weak to measure by FP and ITC. These observations are consistent with other Med15-dependent transcription factors (e.g., Gcn4, Ino2, and Met4)[32,36], whose ADs also display micromolar affinities for Med15 and enhanced affinity when multiple ABDs are present (e.g. Gcn4 ADs in Table 1 and ref. [36]). As well, the rank order of binding strengths for Med15 ABDs to the Gal4 AD and Gcn4 AD are the same, with the strongest binding to ABD1 and the weakest binding to ABD2, suggesting that the differences in affinity are intrinsic to each ABD. Furthermore, we note that the highest affinity Gal4–Med15 interaction is still ~120-fold lower affinity than the binding of Gal4 to Gal80 repressor[6].

**Identification of Gal4 AD residues that interact with Med15 ABD1**. We used NMR to investigate the mechanism of Gal4–Med15 binding and the role of the Gal4 hydrophobic patches. The Gal4 828–881 polypeptide is intrinsically disordered as evidenced by its narrow $^1$H dispersion in the ($^1$H,$^{15}$N)-HSQC (heteronuclear single quantum coherence) NMR spectrum (black peaks, Fig. 3a) and low, uniform $T_1/T_2$ values (green line, Fig. 3c). Despite the narrow spectral dispersion, the NMR spectrum of Gal4 AD could be assigned using conventional triple-resonance protocols. The assignments provided further confirmation that Gal4 AD is disordered, as the backbone $^{13}$C chemical shifts are close to those predicted for random coil (green bars, Fig. 3d).

### Table 1 Affinity of ADs for Med15 activator-binding domains.

**Binding data by FP or (ITC)**

| Med15 | AD | $K_d$ (μM) | Reference |
|---|---|---|---|
| ABD1 | Gal4 AD | 9.24 ± 1.35 (9.2 ± 0.2) | |
| | Gcn4 nAD | 3.3 ± 0.5 | 29 |
| | Gcn4 cAD | 2.5 ± 0.4 | 29 |
| | Gcn4 tAD | 2.66 ± 0.22 | 32 |
| ABD2 | Gal4 AD[a] | 98 ± 15 (35 ± 6.1) | |
| | Gcn4 nAD | 21.8 ± 3.8 | 29 |
| | Gcn4 cAD | 147 ± 30 | 29 |
| | Gcn4 tAD | 19.5 ± 2.74 | 32 |
| ABD3 | Gal4 AD | 16.2 ± 1.0 | |
| | Gcn4 nAD | 2.6 ± 0.4 | 29 |
| | Gcn4 cAD | 13.9 ± 0.5 | 32 |
| | Gcn4 tAD | 4.40 ± 0.49 | 32 |
| KIX | Gal4 AD | Unable to measure | |
| | Gcn4 nAD | Unable to measure | 29 |
| | Gcn4 cAD | Unable to measure | 29 |
| | Gcn4 tAD | Unable to measure | 32 |
| ABD123 | Gal4 AD | 2.04 ± 0.22 | |
| | Gcn4 nAD | 0.36 ± 0.02 | 29 |
| | Gcn4 cAD | 1.7 ± 0.1 | 29 |
| | Gcn4 tAD | 0.109 ± 0.012 | 32 |

Gcn4 ADs are N-terminal (nAD, residues 1–100), central (cAD, residues 101–134), or tandem (tAD, residues 1–134).
[a]The Gal4 AD:ABD2 $K_d$ determined by ITC is used for saturation calculations and is consistent with estimates from NMR titrations.

We characterized the effects of Med15 ABD1 and ABD2 on Gal4 AD. A change in NMR peak positions (chemical shift perturbations (CSPs)) indicates a change in chemical environment of the group that gives rise to the NMR peak. As shown in Fig. 3a, addition of non-isotopically labeled Med15 ABD1 results in large changes in peak position (CSP) for some peaks and no perturbation to others. Affected peaks show continuous, fast-exchange CSPs as a function of increasing Med15 ABD1. The fast exchange behavior and large CSPs provide an estimate of the exchange rate to be at least 250 s$^{-1}$ and therefore a lifetime for the bound state of no more than 4 ms consistent with the micromolar affinity measured for this interaction. We find similar results with Med15 ABD2 (Supplementary Fig. 2). Plotting CSPs along the Gal4 AD sequence reveals substantial perturbations to Gal4 residues spanning from 840 to 870, while the N- and C-termini of the Gal4 AD exhibit only very small CSPs (Fig. 3b) and unchanged dynamics ($T_1/T_2$) (Fig. 3c). Thus, the minimal AD as identified from activity assays (Fig. 2) is congruent with the minimal region affected by both ABD1 and ABD2 according to NMR measurements. The largest perturbations are within Region 2, specifically, residues 861–869. Analysis of the backbone chemical shifts for the Gal4 AD:ABD1 and AD:ABD2 complexes indicate that residues 861–869 adopt helical structure upon binding to both ABD1 and ABD2 (Fig. 3d). This region also shows the largest increase in $T_1/T_2$ dynamics, which is an approximate measure of molecular tumbling time and thus of protein complex size; regions of AD that spend more time bound to ABD1 or ABD2 are expected to have elevated $T_1/T_2$ values. Residues within Region 1 also experience CSPs and small but significant increases in $T_1/T_2$ values, consistent with the region's contribution to Gal4 AD function. Secondary structure prediction based on backbone chemical shifts indicate that Region 1 residues Gln843–Phe850 adopt a modest amount of helical structure. Altogether, the data are consistent with transient interactions between ABD1 or ABD2 and Gal4 that are dominated by Gal4 residues 861–869 with lesser contribution from Region 1.

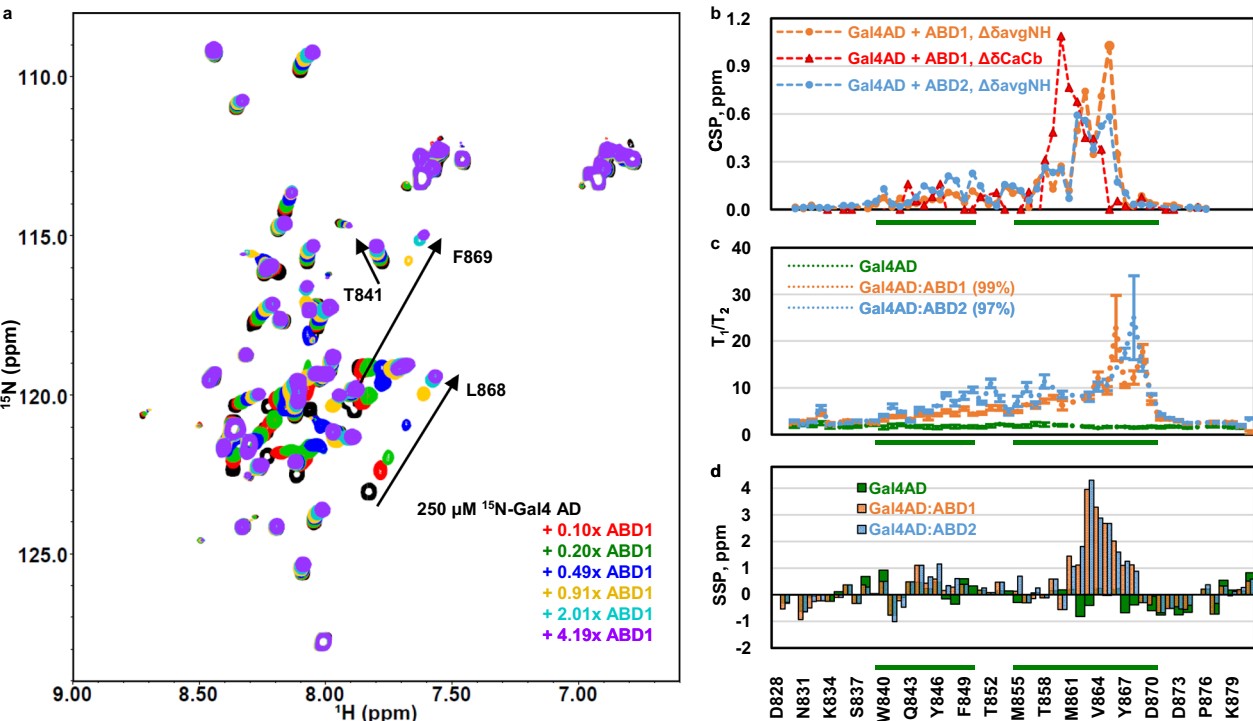

**Fig. 3 Gal4 AD has a primary region of ABD1/2 interaction with smaller effects throughout the N-terminus. a** ($^{1}$H,$^{15}$N)-HSQC titration spectra of $^{15}$N-Gal4 AD with ABD1. Fraction saturation based on the $K_d$ are 0 (black), 0.1 (red), 0.19 (green), 0.46 (blue), 0.78 (yellow), 0.98 (cyan), and 0.99 (purple). **b** Average amide chemical shift perturbation (CSP) quantifying the free and bound shifts are shown as orange lines for ABD1 and blue lines for ABD2. The scaled CSPs for $^{13}$Ca-$^{13}$Cb for free vs. bound Gal4AD:ABD1 are shown as red lines. **c** Backbone dynamics of Gal4 AD free (green) and bound to ABD1 (orange) or ABD2 (blue). The increase of $T_1/T_2$ in Gal4 AD:ABD1/2 extending from the primary interacting region to the N-term suggests a weaker interacting role for those regions. Errors represent the standard error of the fits of the relaxation experiment intensity data. **d** Secondary structure prediction (SSP) for free and ABD1- or ABD2-bound Gal4 AD. The primary interacting region becomes helical (values towards 4 ppm are consistent with fully helical structure, negative values are indicative of beta strand structure). Green bars along the x-axis correspond to the Gal4 AD hydrophobic regions of Fig. 1a. See also Supplementary Fig. 2.

Notably, the sequence that appears to dominate the interaction with the Med15 ABDs is the most highly conserved among Gal4 species (Supplementary Fig. 1) and overlaps with the Gal80-binding site.

**Sequence-independent Med15 ABD1 and ABD2 interactions with acidic ADs**. To observe the interaction from the perspective of Med15, unlabeled Gal4 AD was titrated into $^{15}$N-ABD1 or $^{15}$N-ABD2. As in the reciprocal titrations described above, NMR peaks shifted in fast exchange with increasing Gal4 AD. Remarkably, the chemical shift trajectories of peaks for both ABD1 and ABD2 upon addition of Gal4 AD were nearly identical to those observed upon titration with the tandem Gcn4 ADs (Fig. 4a and Supplementary Fig. 3). Gal4 and Gcn4 have very different primary sequence, but both have numerous hydrophobic regions set in an acidic, disordered background. It is these characteristics that promote transcription activation[26] and the highly similar perturbations observed point to sequence-independent interactions between the acidic activators and Med15. The highly similar perturbations observed point to sequence-independent interactions between the acidic activators and Med15.

Both ABD1 and ABD2 show widespread CSPs upon addition of AD, as plotted on the respective structures in Fig. 4b. One face of ABD1 contains a shallow hydrophobic groove that serves as the AD-binding site, while binding to ABD2 is distributed over its surface centered on two hydrophobic surface patches. This widespread pattern of CSP is highly similar to that observed with the Gcn4 ADs and is consistent with the formation of fuzzy complexes between Gal4 AD and Med15[31–33].

**Gal4 AD forms a fuzzy complex with Med15 ABD1 and ABD2.** Fuzzy interactions cannot be described by a single orientation between binding partners. To detect the orientation of Gal4 AD when bound to Med15, we inserted paramagnetic spin-labels into the Gal4 AD sequence. Single Cys mutations for spin-label attachment were placed on either side of the most perturbed and highly conserved Gal4 AD region (Fig. 5a). TEMPO spin-label was attached to Gal4 AD T860C, E872C, or T874C and added to $^{13}$C,$^{15}$N-ABD1 or $^{13}$C,$^{15}$N-ABD2. The paramagnetic relaxation enhancement (PRE) was determined by quantifying ($^{1}$H,$^{15}$N)- and ($^{1}$H,$^{13}$C)-HSQC spectra with active or inactive spin-label to determine regions of the ABD in proximity to the spin-label site, as peak intensity is lost as a function of time-averaged distance to the spin-label (Fig. 5b–d and Supplementary Fig. 4a, b). In these plots, the largest PREs correspond to the smallest values (toward 0 on the y-axis) and resonances that do not display a PRE will have a value close to 1.0.

For ABD1, each Gal4 AD spin-label site gives rise to PREs. Remarkably, both the pattern and magnitude of PREs are almost identical for labels attached at residues T860 (green trace) and E872 (blue trace) despite their being 12 residues away from each other (Fig. 5b and Supplementary Fig. 4a). In contrast, a label only two residues removed from E872, namely T874, gave somewhat smaller PREs, despite similar saturation (~60–70%) for the Gal4 AD spin-label samples. The pattern of PREs identify distinct regions along the ABD1 sequence that are approached by the spin-labels; these map to residues that encircle a shallow hydrophobic groove on one side of ABD1, with the backside surface largely unaffected (Fig. 5d). The picture is somewhat

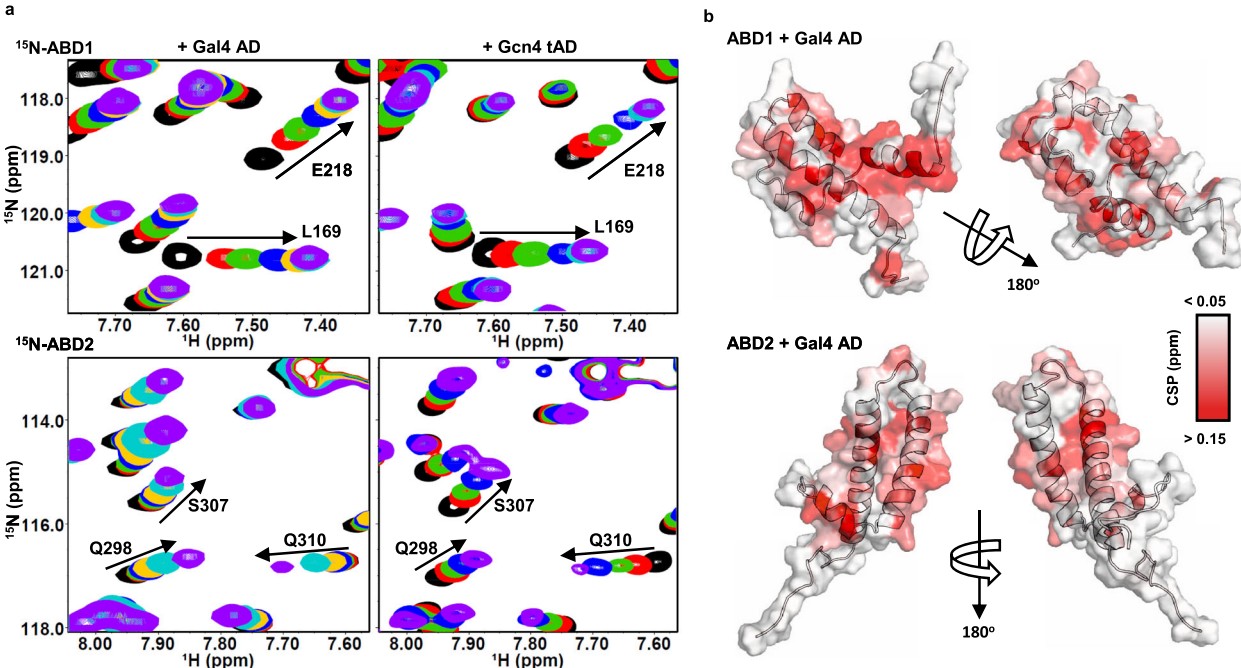

**Fig. 4 Med15 ABDs have widespread CSPs upon addition of Gal4 AD that resemble the Gcn4 tAD interactions. a** ($^1$H,$^{15}$N)-HSQC titration spectra of $^{15}$N-ABD1 (top) or $^{15}$N-ABD2 (bottom) with Gal4 AD (left) or Gcn4 tAD (right). ABD1 and ABD2 show similar chemical shift trajectories regardless of the added activator domain. **b** CSPs of ABD + Gal4 AD are plotted on the structures of ABD1 (pdb 2LPB) and ABD2 (pdb 6ALY) with darker red indicating a larger CSP. See also Supplementary Fig. 3.

different for ABD2. Again, the spin-labels located 12 residues apart produce highly similar PRE patterns and magnitudes (green and blue traces in Fig. 5c and Supplementary Fig. 4b). But rather than distinct regions exhibiting PREs and others exhibiting none, virtually every resonance within the structured regions of ABD2 shows at least some PRE (Fig. 5d). In contrast, the spin-label at position 874, only two residues away, shows little or no effect. For technical reasons, we were unable to achieve identical levels of saturation for the three spin-labeled peptides (~90, 70, and 50% for T860C, E872C, and T874C, respectively). However, this is unlikely to account for the discrepancy, as we have routinely collected our PRE data at around 50% saturation. Instead, the results highlight the difference in binding interfaces of ABD1 vs. ABD2. Gal4 AD T860 is adjacent to the primarily affected residues 861–870 and has small but significant CSPs upon addition of ABD1 (Fig. 3). In contrast, T874 is located within the Gal4 AD C-terminal residues that do not contribute to transcription activity (Fig. 2) nor do they experience significant perturbations in the NMR experiments (Fig. 3). The PREs indicate that Gal4 AD T874 spends more time on average close to ABD1's shallow hydrophobic groove than it does to ABD2, which does not have a clear binding pocket. The most affected ABD2 residues in the T874C spin-label experiment are those that are most affected in the T860C experiment (Q295, I296, N297, V332, and A345) and these residues are far apart in the ABD2 structure. Altogether, our results are consistent with a Gal4 AD C-terminal tail that is oriented away from the ABDs and does not contribute to binding.

The PREs observed from Gal4 AD spin-labels on ABD1 and ABD2 are each consistent with previous results obtained from spin-labeled versions of Gcn4 AD. The pattern of Gal4 AD PREs are highly similar to the pattern of Gcn4 AD PREs, as clearly illustrated in correlation plots (Supplementary Fig. 4c–e). Importantly the analysis reveals that Gal4 AD PREs correlate strongly with each other and with Gcn4 AD PREs. A correlation between PREs that emanate from Gal4 positions that are 12 residues apart

means that these residues, on average, occupy similar positions for similar amounts of time on the ABDs. Thus, the parsimonious conclusion to be drawn is that, as with Gcn4–Med15 binding, a single orientation of bound Gal4 AD cannot satisfy the PRE data. This situation implies that Gal4 AD adopts a fuzzy interface upon binding to Med15 ABD1 and ABD2.

**Multiple ABD:AD structure states are required to satisfy the PRE data.** The highly correlated PREs from multiple spin-label sites is consistent with (but not required for) fuzzy binding. To examine the situation more rigorously, we set out to determine whether PRE data from even a single spin-label site could be satisfied by a fixed orientation ABD:AD structure. The Gal4 AD T860C-TEMPO spin-label data correlate extremely well with the Gcn4 cAD S117C-TEMPO spin-label data previously published[32] (Supplementary Fig. 4c). The NMR solution structure for ABD1 and Gcn4 cAD has been solved (pdb 2LPB)[33] using extensive experimentally derived restraints (chemical shifts, dihedral angles, and nuclear Overhauser enhancements (NOEs); BMRB 18244). We used the structure and associated restraints as a starting point to determine whether the Gcn4 cAD S117C-TEMPO spin-label PRE data could be satisfied with a single-state ABD1:cAD structure or whether instead an ensemble of multiple ABD1:cAD orientations was required. We compared the total energy and PRE energy terms for ensemble structures containing up to six members (see "Methods"). The energy terms calculated for a single-state structure are dramatically improved by addition of even just one additional ensemble member, and additional improvements are predicted for three- and four-member ensembles (Fig. 5e). Ensemble sizes larger than four did not substantially improve the energy terms. The PRE data were converted into an $R_2$-like relaxation rate for structure refinement using Xplor-NIH (black line Fig. 5f). Again, an ensemble of two or more states is required to recapitulate the PRE data (red lines Fig. 5f). The analysis supports the conclusion that ABD1:cAD is a fuzzy complex, as previously determined[33]. By extrapolation,

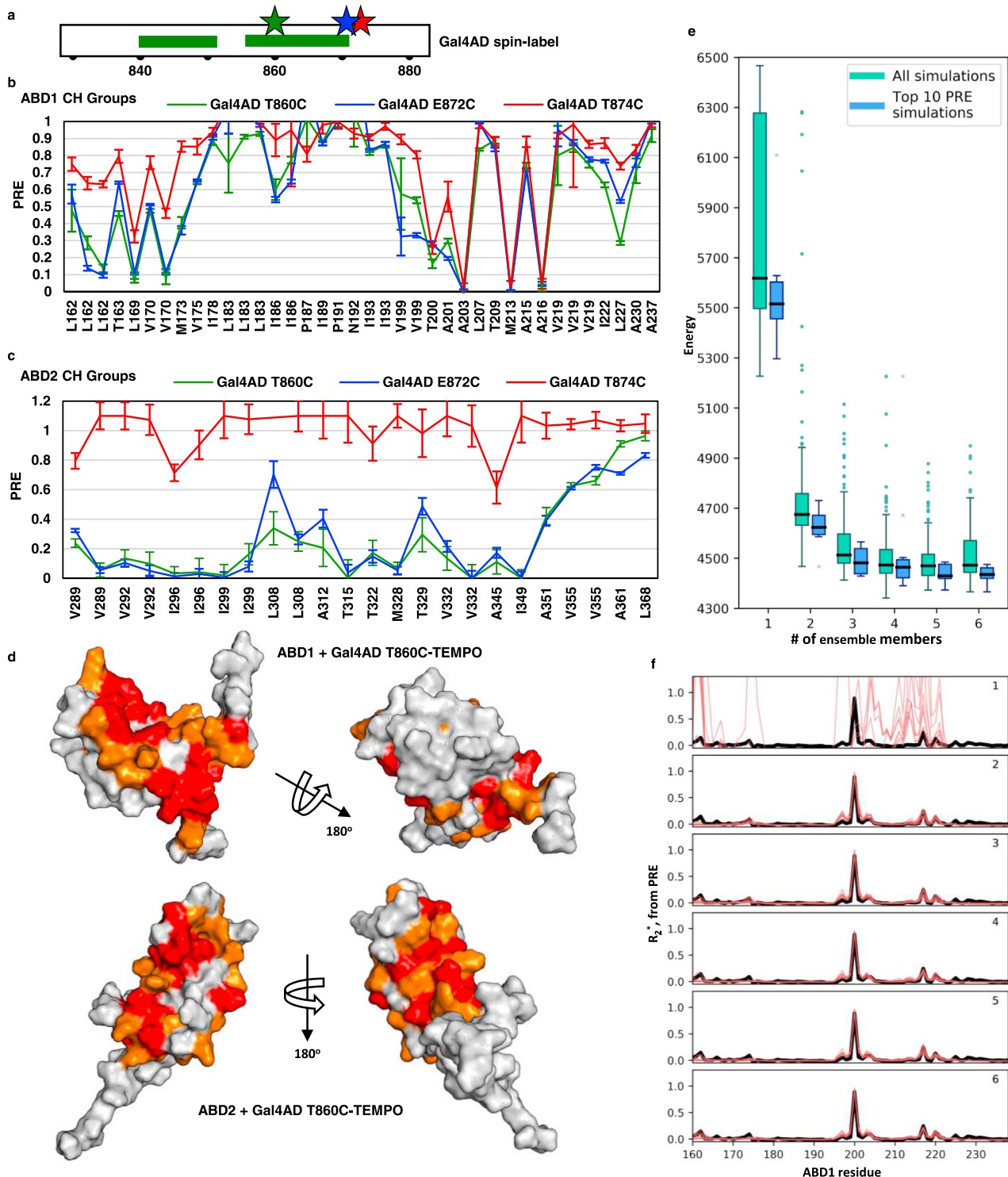

**Fig. 5 Paramagnetic relaxation effects (PRE) on ABD1 and ABD2 when bound to spin-labeled Gal4 AD. a** TEMPO spin-label was attached at Gal4 AD positions T860C, E872C, or T874C. PRE of each spin-labeled AD is shown for CH groups for **b** ABD1 and **c** ABD2 (see Supplementary Fig. 4 for NH group data). Error bars represent standard error-based spectral noise intensities. **d** Results of the ABD + Gal4 AD T860C-TEMPO experiments are plotted on the structures of ABD1 (pdb 2LPB) and ABD2 (pdb 6ALY). The top 25% affected residues are red, the next 25% are orange. A similar PRE pattern of effect is measured when the spin-label is attached at either side of the primary interacting region of Gal4 AD. **e, f** Simulations of the Gcn4 cAD:ABD1 complex using previous data (Brzovic et al.[33], Tuttle et al.[32]) demonstrate that multiple ensemble members are required to satisfy the PRE data. **e** The total energy for all structures (green) and the ten lowest PRE energy term structures decreases significantly when structures have multiple ensemble members. Box plots reflect the 25th to 75th percentile, a dark midline denotes the median, whiskers extend to a maximum of 1.5× IQR beyond the box. **f** The PRE, calculated as a pseudo relaxation rate $R_2^*$ for structure calculations, requires multiple ensemble members to recapitulate the experimental data. Inset number indicates the number of states in each structure: 1 at top to 6 at bottom. See also Supplementary Fig. 4.

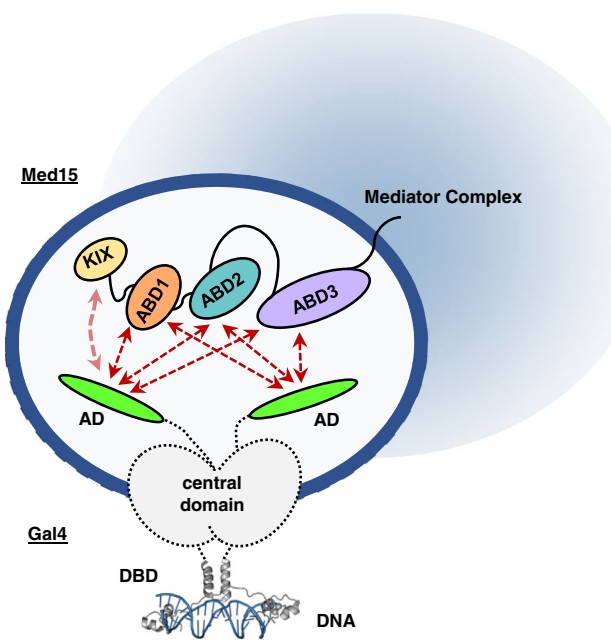

**Fig. 6 Model for interaction of the Gal4 AD and Med15 ABDs in a dynamic fuzzy complex.** Although the Gal4 AD sequence is very different from the Gcn4 ADs and tightly binds Gal80 repressor, the two activators bind Med15 using the same fuzzy free-for-all mechanism. Gal4DBD:DNA structure is from pdb 1D66.

the strong correlation between Gal4 AD PREs and those of Gcn4 cAD supports the conclusion that the Gal4 AD also forms a fuzzy complex with ABD1.

## Discussion

Our functional studies identify two hydrophobic-containing regions in a minimal Gal4 AD that contribute to transcriptional activation. Both regions are required for robust transcription activation in vivo, as each individual region has only modest activation activity on its own. Such behavior is reminiscent of the Gcn4 nAD, in which four hydrophobic clusters distributed over 100 residues are required for full function and no single hydrophobic cluster can activate transcription effectively on its own. Both Gal4 AD regions interact directly with ABDs of Med15, as evidenced by NMR, with Region 2 appearing to dominate the binding. We note that the most C-terminal hydrophobic cluster in the Gcn4 nAD also shows the largest CSPs, despite additional clusters being functionally important. These two examples contrast with the Gcn4 cAD, whose function is largely encoded within a single short sequence embedded in a disordered, acidic polypeptide[31,33,34]. Important hydrophobic residues are spread over a 20–30 residue range in yeast transcription activators Met4 and Ino2, suggesting that hydrophobic residues distributed over an AD are more common than a single short dominant sequence. This model offers a rationale for why conserved sequence motif(s) have not been identified among transcription activators. Furthermore, the interactions are weak and transient, with an upper limit on the lifetime of an AD/ABD1 complex of around 4 ms. We suggest that the multiple short hydrophobic regions contribute to Med15 binding by a mechanism similar to avidity or allovalency[39]. By this mechanism, multiple weak binding sites in the AD act to increase the effective affinity for multiple binding sites on Med15 by decreasing the dissociation rate of the two molecules and effectively increasing the local concentration of the binding partners. This binding mechanism can also explain the

positive contribution of entropy to Med15 binding observed with Gcn4, Ino2, Met4, and Gal4 ADs[31,33,36].

The NMR results confirm that the Gal4 AD is disordered in its unbound state and that part of Region 2 (residues 861–868) adopts helical structure upon binding (Fig. 3d). Yet despite adopting structure, this hydrophobic region binds to ABD1 and ABD2 without adopting a fixed orientation. The two Med15 ABDs investigated in this study are each structured on their own and present preformed binding surfaces. The remarkable similarities of the CSPs induced by Gal4 AD and Gcn4 tAD, which have very different sequences, on each of the ABDs indicate that the two ADs interact similarly with the ABD surfaces, via a hydrophobic cloud rather than via defined side chain-to-side chain interactions. Such sequence-independent effects, along with the lack of a fixed binding orientation, are hallmarks of fuzzy binding. Transient, dynamic interactions of multiple regions of Gal4 AD with each of the Med15 ABDs is consistent with a fuzzy free-for-all mechanism (Fig. 6)[32].

Our results lead to two general conclusions. First, we propose that the fuzzy free-for-all mechanism is common among transcription factors with acidic ADs. This conclusion is based on the demonstrated similarities between the ways in which Gal4 and Gcn4 engage Med15, despite their lack of sequence conservation and the very different mode of regulation of Gal4 and Gcn4 transcription factors. The notion of a common mode of action among ADs is non-trivial. Although the ADs of different transcription factors and DNA-binding domains can be interchanged to yield chimeric proteins of similar function[40–42], the intrinsically disordered ADs have experienced independent evolutionary trajectories that are in part shaped by interacting, coevolving partners. For example, the Gal4 AD has evolved to bind Gal80 repressor with high affinity and sequence specificity yet the same sequence is directly involved in sequence-independent fuzzy interaction with Med15. In contrast, Gcn4 is under no analogous constraint. As intrinsically disordered regions tend to experience relaxed purifying selection (i.e., fewer mutations are highly deleterious to protein function and can accumulate by genetic drift)[43], they also tend to experience stronger positive directional selection from structured coevolving partners who are generally more mutationally constrained. This type of coevolutionary asymmetry between structured and disordered binding partners has been well documented in a pair of rapidly evolving sperm proteins where, perhaps coincidentally, the intrinsically disordered protein is also highly acidic with a hydrophobic patch that interacts with the structured partner[44].

A second conclusion that can be drawn from our study—and may arise from such coevolutionary asymmetry—is that the mode of binding of intrinsically disordered ADs is dictated predominantly by the binding partner, rather than by the disordered component. This conclusion is supported by two observations. First, the binding preferences/strengths of the ADs from Gal4 and Gcn4 to the three ABDs of Med15 follow the same trend, with highest affinity for ABD1 and lowest affinity for ABD2. What is in common are the ABD structures, not the ADs. Second, within the Gal4 AD, a sequence contained within Region 2 binds as a fuzzy complex to Med15 ABDs and binds as a well-oriented, structured polypeptide to the Gal80 repressor[20,21]. This is strong evidence that it is the binding partner that dictates whether a disordered ligand will (1) remain disordered while bound, (2) adopt structure but bind in a fuzzy manner, or (3) bind in a canonical, orientationally defined manner.

Our work has focused on determining what constitutes an activator and we have sought to answer this question for yeast Med15-dependent acidic activators[26,29,31–33,36]. We find that Med15 ABDs recognize the hydrophobic character of acidic activators more than specific sequence and that the AD–ABD

interactions occur via a fuzzy interface. A recent study of different ETV family ADs binding to Med25 ABD revealed three-state binding kinetics[45]. In this case, conformational sampling of the three states depended on sequence, leading the authors to conclude that the ETV ADs bind as "dynamic complexes with several well-populated conformational substates at equilibrium." This description is congruent with fuzzy binding as was determined for other AD-Med25 ABD complexes[46]. Why different transcription factor ADs do vary in sequence is a challenging and not yet answered question. We have previously shown that binding affinity is not a primary selective force as synthetic ADs can be designed that bind ABDs more tightly and activate more strongly than the wild-type ADs[31]. For Gal4 AD, there is evidence that the AD sequence is constrained by the interaction with Gal80 more than by interactions with Med15 ABDs, and it may be that other ADs are similarly constrained by factors other than the ABDs.

Questions remain regarding the overall mechanism of transcription activation. For systems with weak dynamic interactions, condensate formation is proposed to help compartmentalize the components. Compartmentalization could serve both to increase local concentrations of the relevant components to drive the frequency of these interactions as well as to decrease the frequency of non-productive encounters[47]. Recent studies demonstrate the importance of condensate formation for transcription activation via super enhancers, with the AD appearing to be an important driving force for condensate formation[48–51]. A fuzzy free-for-all binding mechanism can be consistent with the formation of condensates[52]. Furthermore, the multiple transcription factor-binding sites that often occur at promoter regions can contribute to higher local concentrations. The *HIS3* promoter has seven Gcn4 consensus binding regions; *HIS5* promoter contains five potential sites for Gcn4[53]; and the promoter controlling *GAL1*, *GAL4*, and *GAL10* transcription contains four Gal4 DBD sites to which Gal4 binds cooperatively[5]. Each DNA site binds a dimer of the relevant transcription factor, presenting two ADs, each of which contain multiple hydrophobic-binding regions per site. In turn, the ADs may simultaneously interact with components of Mediator or other coactivator-binding partners via a fuzzy free-for-all mechanism to promote transcription activation (Fig. 6). It has not yet been established whether condensate formation is a direct consequence of such interactions and whether such formations are functionally important. Notwithstanding, ADs have been implicated in the formation of condensates and there may be further dependence on the Q-rich linker regions of the coactivators[37,48,49,54].

Taken together, our conclusions indicate that transcription activation by Gal4 and Gcn4 with Med15 occurs via a common interaction mechanism that is dictated by the properties of the Med15 ABDs. A remaining question is whether these same ADs use a fuzzy-binding mechanism in their interactions with other coactivators such as with subunits of SAGA or whether the Med15 ABDs are uniquely configured to facilitate fuzzy interactions.

## Methods

**Yeast strains and plasmids**. Yeast strain SHY944 (*matα ade2Δ::hisG his3Δ200 leu2Δ0 lys2Δ0 met15Δ0 trp1Δ63 ura3Δ0 gcn4Δ::KanMx gal80Δ::HIS3*) was transformed with the indicated Gcn4- or Gal4AD-Gcn4-containing plasmids that are derivatives of the *LEU2*-containing vector pRS315. All Gcn4 derivatives contained a C-terminal 3X-Flag tag. All primers used for this work are listed in Supplementary Table 1.

**Total RNA isolation**. Cell cultures were grown in duplicate to an OD600 of 0.8–1.0 in 2% dextrose synthetic complete medium lacking Ile, Val, and Leu at 30 °C. Cells were induced with 0.5 μg/ml SM (Sigma) for 90 min to induce amino acid starvation, and approximately 10 ml of cells were pelleted by centrifugation and washed in cold water. Cells were incubated at 65 °C for 1 h in equal volumes of TES

(10 mM Tris [pH 7.5], 10 mM EDTA, 0.5% sodium dodecyl sulfate) and acid phenol (Ambion). Cells were extracted twice with acid phenol and once with chloroform (Sigma), and RNA was isolated by ethanol precipitation. Fifteen micrograms of RNA were treated with the Turbo DNase Kit (Invitrogen), and 1 μg of DNA-free RNA was used to generate cDNA using Transcriptor (Roche), anchored oligo(dT)18 primer, and the manufacturer's instructions. cDNA was diluted 1:20 for quantitative PCR (qPCR).

**Reverse transcription qPCR**. Gene-specific qPCR was performed in triplicate using primers near the 3′ end of the gene. Primers were designed by either the PrimerQuest (IDT) or the Primer Express 3 (ABI) software using default parameters. qPCRs were assembled in 5-μl reaction mixtures in a 384-well plate format using a QuantStudio5 Real Time PCR System (ABI) and Power SYBR green master mix (ABI). Relative amounts of DNA were calculated using a standard curve generated from tenfold serial dilutions of purified genomic DNA ranging from 10 to 0.001 ng. Data are given in Fig. 2 as mRNA ratios of ARG3 to ACT1 or of HIS4 to ACT1. All values are expressed relative to that of Gal4 828–881, which was set at 1.0.

**Protein purification**. All proteins were expressed in BL21 (DE3) RIL *Escherichia coli*. Med15 484–651 (ABD3) and Med15 158–651 Δ239–272, Δ373–483 (ABD123) were expressed as N-terminal His6-tagged proteins. All other Med15 and Gal4 constructs were expressed as N-terminal His6-SUMO-tagged proteins. Cells were lysed in 50 mM 4-(2-hydroxyethyl)-1-piperazineethanesulfonic acid (HEPES) pH 7.0, 500 mM NaCl, 40 mM Imidazole, 10% glycerol, 1 mM phenylmethanesulfonylfluoride (PMSF), and 5 mM dithiothreitol (DTT) and purified using Ni-Sepharose High Performance resin (GE Healthcare). Proteins were eluted in 50 mM HEPES pH 7.0, 500 mM NaCl, 500 mM Imidazole, 10% glycerol, 1 mM PMSF, and 1 mM DTT. Purified SUMO-tagged proteins were concentrated using 10K molecular weight cutoff centrifugal filters (Millipore); diluted 10× in 50 mM HEPES pH 7.0, 500 mM NaCl, 40 mM Imidazole, 10% glycerol, 1 mM PMSF, and 5 mM DTT; and digested with SUMO protease for 3–5 h at room temperature using ~1:800 protease:protein ratio. Cleaved His6-Sumo tag was removed using Ni-Sepharose. Med15 polypeptides were further purified using HiTrap Heparin (GE Healthcare) in 20 mM HEPES pH 7, 1 mM DTT, and 0.5 mM PMSF eluting with either a 50–350 mM NaCl gradient (ABD1 and ABD2) or a 200–600 mM NaCl gradient (ABD3 and ABD123). Gal4 AD was further purified by chromatography on Source 15Q (GE Healthcare) using a 50–350 mM NaCl gradient. All proteins were further purified using size exclusion chromatography on Superdex 75 10/30 (GE Healthcare). Proteins used in FP and ITC were eluted in 20 mM KH2PO4 pH 7.5, and 200 mM KCl. Proteins used in NMR were eluted in 20 mM NaH2PO4 pH 6.5, 200 mM NaCl, 0.1 mM EDTA, 0.1 mM PMSF, and 5 mM DTT. The concentration of the purified proteins was determined by ultraviolet/visible spectroscopy with extinction coefficients calculated with ProtParam[55].

**FP and ITC binding experiments**. Gal4 peptides used in FP were labeled with Oregon Green 488 dye (Invitrogen) as previously described[29]. FP measurements were conducted using a Beacon 2000 instrument as previously described[29]. Titrations between Gal4 AD and Med15 were performed with 15 concentrations of Med15 spanning 0–250 μM (KIX, ABD1, ABD2, ABD3, ABD123) at 22 °C. All Gal4 AD vs. Med15 titrations were done in triplicate except for ABD123, which was done in duplicate. FP data were analyzed using Prism 7 (Graphpad Software, Inc.) to perform non-linear regression analysis using the one-site total binding model $Y = B_{max} \times X/(K_d + X) + NS \times X + Background$ where $Y$ equals arbitrary polarization units and $X$ equals Med15 concentration.

ITC titrations were performed using a Microcal ITC200 Microcalorimeter in 20 mM KH2PO4 pH 7.5 and 200 mM KCl as described in ref. 33. The following protein concentrations were used, with the syringe molecule listed first and the cell molecule listed second: Gal4 AD (1.11 mM) vs. Med15 ABD1 (0.102 mM); Gal4 AD (1.11 mM) vs. Med15 ABD2 (0.11 mM). The injection settings were 15 × 2.55 μl with 180 s spacing, 22 °C cell temp., Ref power = 10 μcal/s, and 1000 rpm stir speed. Calorimetric data were plotted and fit with a single binding site model using the Origin 7.0 software (Microcal).

**NMR experiments and resonance assignments**. NMR HSQC titration, dynamics, and spin-label experiments were completed on a Bruker 500 MHz Avance III spectrometer. All spectra were collected at 25 °C in NMR buffer (20 mM sodium phosphate, 150 mM NaCl, 0.1 mM EDTA, 0.1 mM PMSF, and 5 mM DTT, pH 6.5) with 7% D2O, unless otherwise specified. Spin-label samples were in NMR buffer but with no DTT.

(1H,15N)- and (1H,13C)-HSQC titration experiments were completed for Gal4-AD:Med15-ABD combinations by adding unlabeled Gal4 AD or ABD (ABD1 or ABD2) to a ~200 μM [13C,15N]-AD or ABD sample, maintaining a constant concentration of the labeled species. $T_1$ and $T_2$ 15N-Trosy HSQC experiments for 200 μM [13C,15N]-Gal4 AD, 200 μM [13C,15N]-Gal4 AD + 640 μM ABD1, and 250 μM [13C,15N]-Gal4 AD + 760 μM ABD2 were collected in an interleaved manner with 8 points each[56]. $T_1$ delays were 10, 40, 80, 120, 160, 320, 640, and 1000 ms; $T_2$ cpmg loop delays were 8.48, 16.96, 25.44, 33.92, 42.40, 50.88, 59.36, and 67.84 ms.

$T_1$ and $T_2$ for each residue were fitted to a single exponential with standard errors reflecting the quality in the fit.

PRE experiments were performed as for Gcn4 and ABD1/2[32]. The spin-label 4-(2-Iodoacetamido)-TEMPO was incorporated at a single Cys mutant for Gal4 AD T860C, E872C, and T874C by incubating the AD Cys mutants with 10× TEMPO overnight at room temperature, followed by several hours at 30 °C. Excess TEMPO was removed by elution over a Nap-10 column and buffer exchange during concentration. $^{15}$N- and $^{13}$C-HSQC spectra of 250 μM [$^{13}$C,$^{15}$N]-ABD1 and 200 μM [$^{13}$C,$^{15}$N]-ABD2 with spin-labeled Gal4 ADs were collected in the presence (reference intensity, $I_{dia}$) and absence of 3 mM ascorbic acid ($I_{para}$). Spin-label Gal4 AD concentrations were 180 μM for T860C with ABD1, 170 μM for E872C with ABD1, 200 μM T874C with ABD1, 650 μM T860C with ABD2, 650 μM E872C with ABD2, and 150 μM for Gal4 T874C with ABD2.

Backbone chemical shift assignments were determined from standard backbone triple-resonance experiments (HNCA, HNCOCA, HNCOCACB, HNCACB, and HNCO) obtained on a 560 μM [$^{13}$C,$^{15}$N]-Gal4 AD sample, a 200 μM [$^{13}$C,$^{15}$N]-Gal4 AD + 640 μM ABD1 sample, and a 250 μM [$^{13}$C,$^{15}$N]-Gal4 AD + 1500 μM ABD2 using a Bruker 600 MHz Avance spectrometer with cryoprobe.

The NH CSP was calculated according to $\Delta\delta_{NH}$ (ppm) = sqrt [$\Delta\delta_H^2 + (\Delta\delta_N/5)^2$]. Component $^1$H and $^{15}$N chemical shift differences are (free − bound) in ppm. The CaCb CSP was calculated according to $\Delta\delta_{CaCb}$ (ppm) = 0.25 × [(Cα-Cβ)$_{free}$ − (Cα-Cβ)$_{bound}$]. Secondary structure propensity was determined from $\Delta\delta$(Cα-Cβ) = (Cα-Cβ)$_{measured}$ − (Cα-Cβ)$_{random coil}$, where the random coil shifts were generated from Gal4 AD sequence using a webserver (https://spin.niddk.nih.gov/bax/nmrserver/Poulsen_rc_CS/)[57].

All NMR spectra were collected using Bruker Topspin 3.2, processed using NMRPipe 8.9[58], and analyzed in nmrViewJ 9.0.2[59]. NMR peak intensities for spin-label experiments were quantified using NMRviewJ and error bars reflect noise levels of the spectra. Percent saturation was calculated based on the $K_d$ for each A:B complex according to %sat = [AB]/A = (1/A) × (C/2 + (C$^2$ − 4 × A × B)$^{0.5}$), where A = [$A_{total}$], B = [$B_{total}$], and C = A + B + K$_d$. Protein structure figures were generated using Pymol 1.8[60].

**PRE agreement with ensemble structure simulations.** Each of the 13 states of the Med15-ABD1:Gcn4-cAD solution structure (pdb 2LPB) was used as a starting point to generate 10 structures each for a single state structure up to ensembles containing 6 states (resulting in 130 structures containing 1–6 states each). Xplor-NIH version 3.0-rc3 was used to generate the models using simulated annealing from 4000K to 25K with torsion dynamics followed by Cartesian minimization[61]. Energy minimization was performed using restraints from NOEs, chemical shift-generated dihedral angels, and the Gcn4 cAD S117C-TEMPO spin-label amide PREs[32,33] (BMRB 18244). PREs were converted into an $R_2$-like relaxation term where $R_2^* = 1/I_{para} − 1/I_{dia}$, from the intensities in the spin-label NMR experiments as described in the previous section. Box plots in Fig. 5e are drawn with boxes reflecting the 25th to 75th percentile, a dark midline denoting the median, and outliers identified as values outside the median ± 1.5 × interquartile range. Figures and data analysis were completed using Jupyter Notebook 6.0.3[62].

**Reporting summary.** Further information on research design is available in the Nature Research Reporting Summary linked to this article.

## Data availability
NMR chemical shifts have been deposited to the BioMagResBank: Gal4 AD 828–881 (BMRB accession number 50086). Other data are available from the corresponding authors upon reasonable request. Source data are provided with this paper.

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

## Acknowledgements

We thank all members of the Klevit and Hahn laboratories for their suggestions and comments during this work. L.M.T. would particularly like to thank past and present women mentors for providing a rigorous and supportive scientific environment. This work was funded by NIH RO1 GM075114 to S.H. and R.E.K.

## Author contributions

L.M.T., S.H., and R.E.K. designed the research. L.M.T., D.P., L.W., D.B.W., and S.H. performed the experiments. L.M.T., D.P., L.W., D.B.W., S.H., and R.E.K. analyzed the data. L.M.T., D.P., D.B.W., S.H., and R.E.K. wrote the paper.

## Competing interests

The authors declare no competing interests.
