## [Peer Review File · Nature Communications]

Reviewers' comments:

Reviewer #1 (Remarks to the Author):

This manuscript reports a study of the interaction between transcription activator Gal4 and Mediator subunit Med15. The main regions for activation function of Gal4 have been identified. It was shown that Gal4-AD binds all three ABDs of Med15 with different affinities, and the binding fashion is very similar to that of Gcn4. This study is parallel to the study of Gcn4-Med15, and the main finding is that ABDs of Med15 can form fuzzy interactions with ADs in a sequence-independent manner. Although the paper has interesting contribution, it does not provide enough broad, fundamental insight for the wide-readership of the journal. In addition, I found that some of the conclusions are not sufficiently supported by the experiments. For example,

1. The PRE patterns of the two spin-labeled proteins (AD-T860C and AD-T874C) are quite different, especially for ABD2. The authors explained that T874 might be out of the binding interface, but if this is the case the data still could not prove that there is not a preferred orientation in binding. For ABD1, a more careful examination of the correlation between the two PRE patterns is needed, such as the scatter plot of intensity loss in the study of Gcn4-Med15 interaction (Fig. S6). Furthermore, the authors should present the results of control experiment of ABD + free TEMP.

2. Although the binding of Gal4-AD to ABD2 and ABD3 are weaker than to ABD1, it is still important to identify the Gal4 AD residues that interact with ABD2 and ABD3 since this would provide holistic information of the fuzzy interaction.

Reviewer #2 (Remarks to the Author):

The manuscript by Tuttle et al. combines different and complementary techniques to characterize the interaction between the intrinsically disordered activation domain of Gal4 and the Med15 subunit of the transcription coactivator, Mediator. The study demonstrates that Gal4 binds Med15 with no orientational preference. Binding is described as a highly dynamic sequence-independent “fuzzy” interaction analogous to their previously published work on the Gcn4-Med15 complex. Findings that the activation domains of Gcn4 and Gal4, which have dissimilar primary sequences, use an identical binding mechanism leads the authors to conclude that the “fuzzy” binding mechanism may be common among transcription factors with acidic activation domains.

Overall, the experiments are well-designed and the data are of high quality. The study lends novel insight into acidic transcription factors and how they target partner coactivators.

I have some minor points and questions for the authors:

1. Line 132-133: “Consistent with the importance of hydrophobicity, substitution of the three aromatic residues in Region 1 to alanine is nearly equivalent to a construct lacking Region 1 (“m4” in Fig. 2)”. The authors do not comment on the T841A substitution. Is this substitution known to have no effect on transcription activation?

2. Line 139: “Individual alanine substitution of residues in Region 2 have differential effects” Are the authors referring to substitutions in the 865YNYLF Region? If so, then the sentence should be more specific. The authors also state that the Y867A or Y865A substitutions have only modest effects on transcription activation. The ARG3 error bar for Y867A (Figure 2A) suggest a range of 50-91% transcription activity. A reduction of 50% transcription activity is not modest. Figure 2B should also show the errors.

3. Line 156: “Binding is enhanced in the presence of all three ABDs”. It is unclear why the authors used a construct in which the three Med15 ABDs are connected by shortened linkers. Linkers are known to affect avidity by modulating the effective concentration of intramolecular binding. The authors need to show that the shortened linkers do not contribute significantly to the reported binding

enhancement.

4. In the spin-labeled NMR experiments, the authors report that the ABD2/Gal4 AD T860C spin-labeled sample was more saturated than the T874C spin-label sample (~90% versus ~50% saturation, respectively). This difference in saturation is significant and presents a major discrepancy in the interpretation of the results. This discrepancy should be resolved by using spin-labeled samples with similar saturation.

5. The conclusion that “the structured binding partner of an intrinsically disordered protein dictates the type of protein interaction” is premature. The authors need to consider the fact that unstructured binding partners may use the same binding mechanism as the structured binding partners reported in the study.

6. Fig. 3D and S2: overlapping bars are hard to read

7. Figure 5: Colors are indistinguishable

8. A Figure showing that Gal4 and Gcn4 have dissimilar sequences would be nice!

Reviewer #3 (Remarks to the Author):

General critique: The stated conclusions of this study are both interesting and important in providing evidence that the ADs of two different acidic activators, Gal4 and Gcn4, both interact with the ABDs in Med15 by a similar mode of multiple, low-affinity interactions without orientation preference, indicative of a fuzzy-free-for-all mechanism. In general, however, I find the presentation and description of the NMR data to be insufficient to convince the reader that the Gcn4 and Gal4 ADs interact with the Med15 ABDs 1 and 2 in highly similar ways despite their distinct amino acid sequences. Evidence for strong overlap of the interacting residues for each AD within ABDs 1 and 2 could be better supported by comparing the surface distributions of the CSPs on the two ABDs for Gal4 vs. Gcn4 ADs, which should appear very similar according to the authors conclusions. Even more evidence and a much clearer exposition of the results from the spin labeling approach is needed to convince the non-NMR expert that the Gal4 and Gcn4 ADs both interact with ABD1 without orientation preference, establishing a hallmark of fuzzy interactions. It is left unstated whether the AD interactions with ABD2 also show a lack of orientation preference for either Gal4 or Gcn4; and if this is not the case, then do the AD/ABD2 interactions not conform to a fuzzy mode? The evidence also needs to be more clearly stated to justify the claim that the sequence in Gal4 AD2 shown previously to bind to Gal80 in a defined fashion binds to the Med15 ABDs via fuzzy, ie. orientation independent, fashion, as it is not obvious that this conclusion can be made for a specific sequence withing the Gal4 AD.

Specific comments:

-line 124, Fig. 2: at odds with text, the 847-881 construct appears identical in function to the larger 840-887 construct.

-line 145: It seems that AD1 and AD2 don't function equally, as AD2 appears to be relatively more important.

-line 163: it's not self-evident that there is stronger than additive binding affinity when ABD1-ABD3 are all present vs. each singly—explain this better.

-Table 1: define abbreviations for Gcn4 ADs: n, c, and t

-line 165: it might be worth noting that this statement is true even though the Gcn4 AD binds more tightly overall than does the Gal4 AD to the Med15 ABDs.

-lines 187-190: somewhat at odds with text, AD2 seems to give much stronger CSPs vs AD1, and the difference does not seem that congruent with the relative functions of AD1 and AD2 in cells.

-lines 208-210: the similarity in CSPs for Gal4 vs Gcn4 ADs are hard to appreciate from from Fig. S2. Would it be possible to show a comparison of how the CSPs map on the surfaces of ABD1 and ABD2 for the Gal4 AD and perhaps the Gcn4 tAD as well? This would also document the key claim in lines

219-221 of a common interaction surface in the ABDs for the Gal4 and Gcn4 ADs

-lines 212-213: reference citations for this statement are required

-lines 237-239: stipulate the ABD1 residues around the hydrophobic groove, and indicate what this effect signifies. Also annotate the location of this groove in Fig. 5F and Fig. S3B.

-lines 243-253: It is exceedingly difficult for a non-specialist to understand the differences in the T860 vs T874 spin label results for ABD1 vs ABD2. The goal of this analysis is to provide evidence for lack of orientation in the AD interactions with the ABDs, but whether the results establish this for both ABD1 and ABD2 is far from clear from the current text that would justify the final conclusion on lines 256-258. Indicate the key results in Fig. 5 that justify the claim on lines 287-288 of the Discussion that the Gal4 AD binds to ABD1 without orientational preference. And do the data indicate orientational preference for ABD2?

-lines 253-256: this similarity is far from self-evident. The Gcn4 results appear to be similar to those for Gal4 for the ABD1 constructs, but very different for the ABD2 constructs, and it's unclear exactly what this difference signifies in physical terms for the two ABDs. Presumably, they want to conclude that the Gcn4 AD binds to ABD1 without orientational preference, but how can this be established with only a single spin label for Gcn4?

-lines 315-316: What is the evidence that this sequence in AD2 of Gal4 binds to the Med15 ABDs as a fuzzy complex, i.e. without orientation preference?

Tuttle et al Nat. Comm. Reviewers' comments:

Reviewer #1 (Remarks to the Author):

This manuscript reports a study of the interaction between transcription activator Gal4 and Mediator subunit Med15. The main regions for activation function of Gal4 have been identified. It was shown that Gal4-AD binds all three ABDs of Med15 with different affinities, and the binding fashion is very similar to that of Gcn4. This study is parallel to the study of Gcn4-Med15, and the main finding is that ABDs of Med15 can form fuzzy interactions with ADs in a sequence-independent manner. Although the paper has interesting contribution, it does not provide enough broad, fundamental insight for the wide-readership of the journal.

Many big questions about how ADs function in transcription regulation remain. Our results are part of the beginnings of understanding what constitutes an activator domain (sequence) and why. The notion of a common mode of action among ADs is non-trivial. In particular, there was no reason to predict *a priori* that Gal4 AD-ABD interactions would adopt the same mode as others given that the AD is sequence restrained by its interaction with Gal80. We have expanded the Discussion section to include text on molecular evolution issues associated with the phenomena observed. In addition, the physical mechanism of fuzzy free-for-all binding between ADs and ABDs may be important for condensate formation—a phenomenon that is increasingly recognized as a feature of transcription regulation. For these reasons, the topics presented in this manuscript are of broad interest.

In addition, I found that some of the conclusions are not sufficiently supported by the experiments. For example,

1. The PRE patterns of the two spin-labeled proteins (AD-T860C and AD-T874C) are quite different, especially for ABD2. The authors explained that T874 might be out of the binding interface, but if this is the case the data still could not prove that there is not a preferred orientation in binding.

To address this question, we performed additional PRE experiments with a spin-label at AD position E872. Although the new site is only two residues away from T874, the resulting data correlate strongly with the T860 spin-label data for both ABD1 and ABD2 (Figures 5 and S4). We propose that the correlation of the PRE data at such distal AD sites is consistent with multiple orientation binding. To provide additional support for this interpretation, we now include structure simulations that show that the PRE data from a single spin-label site cannot be satisfied by a single state structure (Figure 5E-F). The new data and analysis support our original conclusion and add strong support for it.

For ABD1, a more careful examination of the correlation between the two PRE patterns is needed, such as the scatter plot of intensity loss in the study of Gcn4-Med15 interaction (Fig. S6). Furthermore, the authors should present the results of control experiment of ABD + free TEMP.

We have revised the PRE figures for clarity and have added the requested scatter plots (Figures 5 and S4). The free TEMPO control was performed and reported in a previous publication (Tuttle et al., 2018).

2. Although the binding of Gal4-AD to ABD2 and ABD3 are weaker than to ABD1, it is still important to identify the Gal4 AD residues that interact with ABD2 and ABD3 since this would provide holistic information of the fuzzy interaction.

We thank the reviewer for the suggestion and now include the complete data for Gal4 AD + ABD2, which show that the same set of Gal4 AD residues are involved in both ABD1 and ABD2 interactions (Figures 3 and S2).

Reviewer #2 (Remarks to the Author):

The manuscript by Tuttle et al. combines different and complementary techniques to characterize the interaction between the intrinsically disordered activation domain of Gal4 and the Med15 subunit of the transcription coactivator, Mediator. The study demonstrates that Gal4 binds Med15 with no orientational preference. Binding is described as a highly dynamic sequence-independent “fuzzy” interaction analogous to their previously published work on the Gcn4-Med15 complex. Findings that the activation domains of Gcn4 and Gal4, which have dissimilar primary sequences, use an identical binding mechanism leads the authors to conclude that the “fuzzy” binding mechanism may be common among transcription factors with acidic activation domains.

Overall, the experiments are well-designed and the data are of high quality. The study lends novel insight into acidic transcription factors and how they target partner coactivators.

I have some minor points and questions for the authors:

1. Line 132-133: “Consistent with the importance of hydrophobicity, substitution of the three aromatic residues in Region 1 to alanine is nearly equivalent to a construct lacking Region 1 (“m4” in Fig. 2)”. The authors do not comment on the T841A substitution. Is this substitution known to have no effect on transcription activation?

We have measured transcription activity of “m4b” which is the m4 mutation with T841 instead of T841A. The measured activity for m4b is the same as the previously reported m4, showing T841A does not have a significant effect on transcription activation. Figure 2 has been updated accordingly.

2. Line 139: “Individual alanine substitution of residues in Region 2 have differential effects” Are the authors referring to substitutions in the 865YNYLF Region? If so, then the sentence should be more specific. The authors also state that the Y867A or Y865A substitutions have only modest effects on transcription activation. The ARG3 error bar for Y867A (Figure 2A) suggest a range of 50-91% transcription activity. A reduction of 50% transcription activity is not modest. Figure 2B should also show the errors.

We have clarified the text and now provide the RT qPCR results as **Supplemental Table 1**, which includes the results of each replicate, the mean, and standard error.

3. Line 156: “Binding is enhanced in the presence of all three ABDs”. It is unclear why the authors used a construct in which the three Med15 ABDs are connected by shortened linkers. Linkers are known to affect avidity by modulating the effective concentration of intramolecular binding. The

authors need to show that the shortened linkers do not contribute significantly to the reported binding enhancement.

The length of the linkers between the ABDs is variable across yeasts (Cooper and Fassler, 2019; Gallagher et al., 2019). We chose to use shortened linkers to be consistent with a construct used in previous NMR experiments (Tuttle et al., 2018). We expect that linker length and content may affect relative binding affinities when comparing individual ABD binding to the whole, but also note that the full extent of the role of these linkers in vivo is not yet known. We have added these points to the text. We do have unpublished transcription activity results that show that shortening the Q-rich linkers between the ABDs regions in a KIX-ABD123 construct had only a modest (~50%) enhancement of transcription activation from ARG3 and no effect at HIS4.

4. In the spin-labeled NMR experiments, the authors report that the ABD2/Gal4 AD T860C spin-labeled sample was more saturated than the T874C spin-label sample (~90% versus ~50% saturation, respectively). This difference in saturation is significant and presents a major discrepancy in the interpretation of the results. This discrepancy should be resolved by using spin-labeled samples with similar saturation.

Because there is rapid exchange between any interacting sites, it is reasonable to expect that a change in saturation will only change the magnitude of the PRE but not the pattern of affected residues (compared to the case where new interactions occur at higher saturations, in which case the pattern would be expected to be different at high versus low saturation). However, to strengthen our PRE-derived results, we have added AD-E872 spin-label experiments, performed at similar saturation as the AD-T860 experiments. See also our response to Reviewer #1, Point #1 above.

5. The conclusion that “the structured binding partner of an intrinsically disordered protein dictates the type of protein interaction” is premature. The authors need to consider the fact that unstructured binding partners may use the same binding mechanism as the structured binding partners reported in the study.

We have sought to clarify and strengthen this point in the text. We give evidence that support that Med15-dependent acidic ADs bind the Med15 ABDs via a fuzzy interface. One of the key features of these AD-ABD interactions is that the ADs are sequence permissive: transcription activity is broadly robust to single mutations and the sequence requirements are generally that there be large hydrophobic regions embedded in a flexible acidic background (Erijman et al., 2020; Pacheco et al., 2018). Contrast this with the high affinity, fixed orientation binding of Gal4 AD with Gal80, which requires sequence-specific interactions.

6. Fig. 3D and S2: overlapping bars are hard to read

7. Figure 5: Colors are indistinguishable

Figures have been modified for clarity and to include new data.

8. A Figure showing that Gal4 and Gcn4 have dissimilar sequences would be nice!

Now included as Figure S1B.

Reviewer #3 (Remarks to the Author):

General critique: The stated conclusions of this study are both interesting and important in providing evidence that the ADs of two different acidic activators, Gal4 and Gcn4, both interact with the ABDs in Med15 by a similar mode of multiple, low-affinity interactions without orientation preference, indicative of a fuzzy-free-for-all mechanism. In general, however, I find the presentation and description of the NMR data to be insufficient to convince the reader that the Gcn4 and Gal4 ADs interact with the Med15 ABDs 1 and 2 in highly similar ways despite their distinct amino acid sequences. Evidence for strong overlap of the interacting residues for each AD within ABDs 1 and 2 could be better supported by comparing the surface distributions of the CSPs on the two ABDs for Gal4 vs. Gcn4 ADs, which should appear very similar according to the authors conclusions.

We have performed additional experiments to address the points raised. Experiments for Gal4 AD + ABD2 have been completed and show similar interactions as with ABD1 (Figures 3 and S2). CSP data comparing the perturbations to ABD1 and ABD2 with each Gal4 AD and Gcn4 ADs is given in Figure S3. The PRE data from spin-label experiments also supports the similar interaction of ABDs with Gcn4 and Gal4 ADs. Figures 5 and S4 have been revised to better show this.

Even more evidence and a much clearer exposition of the results from the spin labeling approach is needed to convince the non-NMR expert that the Gal4 and Gcn4 ADs both interact with ABD1 without orientation preference, establishing a hallmark of fuzzy interactions. It is left unstated whether the AD interactions with ABD2 also show a lack of orientation preference for either Gal4 or Gcn4; and if this is not the case, then do the AD/ABD2 interactions not conform to a fuzzy mode? The evidence also needs to be more clearly stated to justify the claim that the sequence in Gal4 AD2 shown previously to bind to Gal80 in a defined fashion binds to the Med15 ABDs via fuzzy, ie. orientation independent, fashion, as it is not obvious that this conclusion can be made for a specific sequence withing the Gal4 AD.

We have added additional spin-label experiments and structure simulations to strengthen our conclusion of fuzzy binding for Gal4 AD with the Med15 ABDs (see also remarks to Reviewers #1 and #2). We have also taken more care with our language as to what is meant by fuzzy binding: namely, that the interactions between binding partners cannot be described by a single orientation of one binding partner to the other. Our data show that for Gal4 AD and Gcn4 tAD, binding to Med15 ABDs occurs via a protein-protein complex of multiple exchanging conformational states.

Specific comments:

-line 124, Fig. 2: at odds with text, the 847-881 construct appears identical in function to the larger 840-887 construct.

-line 145: It seems that AD1 and AD2 don't function equally, as AD2 appears to be relatively more important.

-line 163: it's not self-evident that there is stronger than additive binding affinity when ABD1-ABD3 are all present vs. each singly—explain this better.

-Table 1: define abbreviations for Gcn4 ADs: n, c, and t

-line 165: it might be worth noting that this statement is true even though the Gcn4 AD binds more tightly overall than does the Gal4 AD to the Med15 ABDs.

- lines 187-190: somewhat at odds with text, AD2 seems to give much stronger CSPs vs AD1, and the difference does not seem that congruent with the relative functions of AD1 and AD2 in cells.
- lines 208-210: the similarity in CSPs for Gal4 vs Gcn4 ADs are hard to appreciate from from Fig. S2. Would it be possible to show a comparison of how the CSPs map on the surfaces of ABD1 and ABD2 for the Gal4 AD and perhaps the Gcn4 tAD as well? This would also document the key claim in lines 219-221 of a common interaction surface in the ABDs for the Gal4 and Gcn4 ADs
- lines 212-213: reference citations for this statement are required
- lines 237-239: stipulate the ABD1 residues around the hydrophobic groove, and indicate what this effect signifies. Also annotate the location of this groove in Fig. 5F and Fig. S3B.
- lines 243-253: It is exceedingly difficult for a non-specialist to understand the differences in the T860 vs T874 spin label results for ABD1 vs ABD2. The goal of this analysis is to provide evidence for lack of orientation in the AD interactions with the ABDs, but whether the results establish this for both ABD1 and ABD2 is far from clear from the current text that would justify the final conclusion on lines 256-258. Indicate the key results in Fig. 5 that justify the claim on lines 287-288 of the Discussion that the Gal4 AD binds to ABD1 without orientational preference. And do the data indicate orientational preference for ABD2?
- lines 253-256: this similarity is far from self-evident. The Gcn4 results appear to be similar to those for Gal4 for the ABD1 constructs, but very different for the ABD2 constructs, and its unclear exactly what this difference signifies in physical terms for the two ABDs. Presumably, they want to conclude that the Gcn4 AD binds to ABD1 without orientational preference, but how can this be established with only a single spin label for Gcn4?
- lines 315-316: What is the evidence that this sequence in AD2 of Gal4 binds to the Med15 ABDs as a fuzzy complex, ie. without orientation preference?

We have clarified the text throughout the document and addressed specific comments as appropriate. Based on comments from all reviewers, we have performed additional experiments that are now included and we believe greatly strengthen our conclusions.

References

- Cooper, D.G., and Fassler, J.S. (2019). Med15: Glutamine-Rich Mediator Subunit with Potential for Plasticity. *Trends Biochem Sci* 44, 737-751.
- Erijman, A., Kozlowski, L., Sohrabi-Jahromi, S., Fishburn, J., Warfield, L., Schreiber, J., Noble, W.S., Soding, J., and Hahn, S. (2020). A High-Throughput Screen for Transcription Activation Domains Reveals Their Sequence Features and Permits Prediction by Deep Learning. *Mol Cell* 78, 890-902 e896.
- Gallagher, J.E.G., Nassif, C., and Pupo, A. (2019). The polymorphic PolyQ tail protein of the Mediator Complex, Med15, regulates variable response to stress. *bioRxiv*, 652669.
- Pacheco, D., Warfield, L., Brajcich, M., Robbins, H., Luo, J., Ranish, J., and Hahn, S. (2018). Transcription Activation Domains of the Yeast Factors Met4 and Ino2: Tandem Activation Domains with Properties Similar to the Yeast Gcn4 Activator. *Mol Cell Biol* 38.
- Tuttle, L.M., Pacheco, D., Warfield, L., Luo, J., Ranish, J., Hahn, S., and Klevit, R.E. (2018). Gcn4-Mediator Specificity Is Mediated by a Large and Dynamic Fuzzy Protein-Protein Complex. *Cell Rep* 22, 3251-3264.

List of Alterations

Changes to the manuscript have been highlighted in grey in the main text document. These changes include:

- An additional author (Damien B. Wilburn) has been added to the paper for his contributions to the PRE data analysis, ensemble structure simulations, and AD evolution discussion
- Several new experiments were completed, with their corresponding methods, results, and discussion incorporated into the text
 - “m4b” Gal4 AD mutation construct was assayed for transcription activity
 - NMR HSQC titration spectra, T1/T2 experiments, and 3D assignment experiments were collected on ^{13}C , ^{15}N Gal4 AD + ABD2 samples
 - Spin-label experiments were completed for a new site, Gal4 AD E872
 - Ensemble structure simulations were completed for ABD1:cAD
- Figure 2 has been updated to include construct m4b
- Figure 3 has been updated to include data from the new Gal4 AD + ABD2 experiments
- Figure 4 has been updated to include 180° rotations of CSP plots on structures
- Figure 5 has been significantly modified to simplify the presentation of PRE data, include new data from the Gal4 AD E872C-TEMPO spin-label experiments, and new data from ensemble structure simulations
- Figure S1 now includes an alignment of Gal4 AD and Gcn4 cAD sequences
- Figure S2 is new, related to Figure 3, is the Gal4 AD + ABD2 HSQC titration spectra
- Figure S3 was formerly Figure S2
- Content of old Figure S3 is included in new main text Figures 4 and 5.
- Figure S4 is new, related to Figure 5, contains additional PRE data and scatter plots

Reviewers' comments:

Reviewer #1 (Remarks to the Author):

The authors have addressed all my concerns.

Reviewer #2 (Remarks to the Author):

All the points I raised were well-addressed by your revisions

Reviewer #3 (Remarks to the Author):

I am fully satisfied with the revisions of text and new experimental data provided by the authors to address my comments on the previous version of the paper; and, hence, believe it is ready for publication.